# Wavelet-Based Transmissibility for Structural Damage Detection

**DOI:** 10.3390/ma15082722

**Published:** 2022-04-07

**Authors:** Kajetan Dziedziech, Wiesław Jerzy Staszewski, Krzysztof Mendrok, Biswajit Basu

**Affiliations:** 1Department of Robotics and Mechatronics, AGH University of Science and Technology, al. Mickiewicza 30, 30-059 Krakow, Poland; dziedzie@agh.edu.pl (K.D.); mendrok@agh.edu.pl (K.M.); 2Department of Civil, Structural and Environmental Engineering, School of Engineering, Trinity College Dublin, D02 PN40 Dublin, Ireland; basub@tcd.ie

**Keywords:** time-variant systems, wavelet-based transmissibility and coherence, output-only analysis

## Abstract

Short-time, abrupt events—such as earthquakes and other shock loadings—often lead to damage that is difficult to detect in structures using output-only vibration measurements. The time-variant transmissibility is proposed to tackle this problem. The approach is based on two-dimensional wavelet power spectra. The time-frequency transmissibility and relevant coherence function are used for structural damage detection in structural elements in buildings. Numerical simulations and experimental tests are used in these investigations. The results are compared with the classical transmissibility and time-variant input-output wavelet approach. The paper shows that output-only measurements and wavelet-based transmissibility can be used to monitor abrupt damage-related changes to structural dynamics.

## 1. Introduction

Vibration measurements have been widely used for many decades for structural health and condition monitoring applications. Previous work in this field includes methods based on time and frequency domain analysis. The former usually utilize direct vibration response measurements, and simple parameters such as maximum vibration amplitudes, root mean square estimates, kurtosis, or vibration instantaneous characteristics (e.g., envelope function, instantaneous frequency). These methods have been known in condition monitoring applications for more than five decades [1]. The latter involves the analysis based on the Fourier analysis and power spectra. Applications relate mostly to civil engineering and transportation. Various methods—based on vibration/dynamic characteristics—have been developed in this area for structural health monitoring since the pioneering work on the damage-related shift of natural frequency, published in the late 1970s [2]. These methods rely on input (excitation) and output (response) measurements. This includes methods based on modal analysis and parameters/characteristics such as: natural frequencies [2], damping [3], deflection shapes [4], mode shapes and curvatures [5,6,7], flexibilities [8], node positions [9], Frequency Response Functions [10], Modal Assurance Criterion (MAC) [11], modal strain energy [12], and various nonlinear methods [13]. A good overview of all these developments can be found in [14,15,16].

There are three major difficulties associated with vibration-based methods used for damage detection. Firstly, parameters developed for damage detection are often embedded in the background noise and/or corrupted by non-damage-related (e.g., environmental, loading) effects. Such parameters are then not sensitive enough to detect early structural damage. This problem is widely acknowledged, and therefore vibration-based methods are not used in many safety-critical applications (e.g., composite damage detection in aerospace engineering [17]). Secondly, many damage events result in short-time, non-stationary vibration characteristics that are not easy (if possible) to analyze using classical modal approaches. The problem can be overcome by applying time-variant methods, reported, for example, in [18,19,20,21,22]. Thirdly, excitation measurements are not always possible in field applications, and only vibration responses can be used for damage detection. The operational modal analysis offers some help to tackle this problem. Various damage detection methods based on operational modal analysis have been developed over the last few decades [23,24]. In addition, transmissibility—based only on response measurements—can also be used for damage detection to avoid the third problem.

The transmissibility concept is well known for vibration suppression problems [25]. For a Single-Degree-Of-Freedom (SDOF) system, transmissibility is often defined as the ratio of two response spectra. This definition has been extended to Multi-Degree-Of-Freedom (MDOF) systems independently in [26,27,28,29]. A good overview of various concepts and properties of transmissibility can be found in [30]. The method was first proposed for structural damage detection in the early 1990s [31]. The application of transmissibility for damage detection is of great interest in field measurements when excitation measurements are not possible. In contrast to operational modal analysis—which assumes uniform flat excitation over the analyzed frequency range—this damage detection approach does not require any excitation assumption. In contrast to Frequency Response Function (FRF) based approaches, the method is also better suited for local damage detection, as indicated in [32,33]. Other applications in structural health monitoring include research reported in [34,35,36,37,38,39,40,41,42,43,44,45,46,47,48,49,50]. The work in [48,49] is based on wavelet analysis that detects local variation of identified properties. The analytical approach reported in [50] is applied to nonlinear systems.

More recently, wavelet-based transmissibility defined in the combined time-frequency domain—has been proposed [51]. Although the method is based on output-only data and can be used to analyze time-variant and/or nonlinear systems, this time-variant transmissibility has not been applied for structural damage detection. The work presented in this paper attempts to fulfill this gap. The wavelet-based transmissibility defined in the time-frequency domain is applied for structural health monitoring. The motivation behind the research undertaken is whether simple vibration response measurements can be used to detect/monitor non-stationary, short-time, abrupt damage-related events caused, for example, by earthquakes. The major difficulty in this application is that when the abrupt change of stiffness related to structural integrity is not detected instantaneously, the estimated stiffness reduction is often too small to reliably indicate damage in the post-earthquake inspection. This is because baseline measurements—that indicate non-damaged conditions—are often not available. Even if these measurements exist, structural parameter changes could also relate to the well-known structural aging problem and/or environmental effects.

The structure of the paper is as follows. Section 2 briefly introduces the theory of time-variant transmissibility. The two-dimensional, time-frequency transmissibility is introduced through the wavelet-based power spectra. The application of the method for damage detection is illustrated in Section 3. Numerical simulations that involve a time-variant lumped parameter building model are described, and damage detection results are illustrated. The experimental work undertaken to demonstrate the damage detection capability of the proposed method is described in Section 4. Two experimental damage-related examples are demonstrated, i.e., detection of abrupt change of stiffness in a structural element of a frame structure and detection of abrupt change of stiffness in a three-floor building model. The wavelet-based transmissibility results—presented in Section 3 and Section 4—are compared with the results based on the classical transmissibility used to analyze the time-variant system. The wavelet-based time-frequency analysis also serves as a reference for the proposed method. Finally, the paper is concluded in Section 5. The results presented in this paper demonstrate that simple vibration response measurements can be used to produce wavelet-based transmissibility that, in turn, can be applied for difficult damage detection problems.

## 2. Wavelet-Based Vibration Characteristics

This section introduces time-frequency vibration characteristics used for structural damage detection. Two-dimensional time-frequency spectra based on the continuous wavelet transform are defined initially. Then wavelet-based FRF (input-output analysis) and wavelet-based transmissibility (output-only analysis) are introduced. In addition, relevant coherence functions are also defined to accompany the entire analysis.

### 2.1. Wavelet Analysis

Vibration and modal analysis—developed originally for linear and time-invariant systems—is based on the Fourier transform. Power spectra are used to construct the input-output characteristics such as FRFs. When the entire analysis is extended to time-variant systems, the Fourier space is usually replaced by the combined time-frequency space. The work in [20,21] involves the continuous wavelet transform defined as
(1)$$W_{x}\left(a,b\right)=\frac{1}{\sqrt{a}}\int_{-\infty }^{+\infty }x\left(t\right)\psi^{*}\left(\frac{t-b}{a}\right)dt$$
where b is a translation operator indicating a location in time, a is a scale operator indicating a location in frequency, $\psi \left(t\right)$ is a mother wavelet and superscript “*” indicates a complex conjugate. The transformed time signal is given by $x\left(t\right)$. The normalization was done by $1/\sqrt{a}\;$in this equation preserves energy independently of the scaling a. It is well known that the scaling parameter a is directly related to frequency. Various wavelets can be used to make the analyzed frequency bandwidth specific for desired applications. Although the selection of wavelet function impacts the entire analysis, previous studies in [20,21] show that the complex Morlet wavelet function is quite suitable for the combined time-frequency analysis of vibration data. This wavelet—composed of the Gaussian window as an envelope and the complex exponential carrier of frequency $\omega_{0}$—is defined as
(2)$$\psi \left(t\right)=e^{\frac{-{\left|t\right|}^{2}}{2}}e^{i\omega_{0}t}\;$$

It is assumed that the work and results based on the Morlet wavelet function are without any loss of generality.

### 2.2. Wavelet-Based Transmissibility

The time-invariant transmissibility can be defined as the ratio of power spectra of two vibration responses (acceleration, velocity, or displacement) measured in two different locations. When the wavelet transform is used instead of the Fourier transform, the method can be extended to the combined time-frequency domain. It can be used for the analysis of time-variant systems, as demonstrated in [51].

To calculate the wavelet-based input-output and output-only characteristics wavelet-based auto- and cross-power spectra are needed and can be defined as
(3)$$G_{ii}\left(a,b\right)=W_{x_{i}}\left(a,b\right)W_{x_{i}}{}^{*}\left(a,b\right)$$
(4)$$G_{ij}\left(a,b\right)=W_{x_{i}}\left(a,b\right)W_{x_{j}}{}^{*}\left(a,b\right)$$
respectively. For a given $x_{k}$ excitation sand relevant $x_{i}$ response signals Equations (3) and (4) can be used to define the wavelet-based FRF as
(5)$$FRF\left(a,b\right)=\frac{G_{ij}\left(a,b\right)}{G_{kk}\left(a,b\right)}$$

In contrast, for two independent responses $x_{i}$ and $x_{j}$ the wavelet-based transmissibility can be defined as
(6)$$T\left(a,b\right)=\frac{G_{ij}\left(a,b\right)}{G_{jj}\left(a,b\right)}$$

The wavelet-based transmissibility defined by the above Equation is now a two-dimensional function. This function combines a set of classical transmissibilities for all specific values of time. The wavelet-based transmissibility can also be used for MDOF systems. Then the appropriate spectra will become matrices that group these spectra for independent coordinates. Some properties of the wavelet-based direct transmissibility arise directly from Equation (6). The proposed function is independent of the force excitation and can be obtained directly from the response measurements. More discussion is provided in [51].

### 2.3. Wavelet-Based Coherence

The coherence function is used in vibration/modal analysis with FRF characteristics to measure the quality of results. The coherence extends the Pearson’s correlation coefficient and indicates how much of the measured responses relate to excitations. This function is also used in practical engineering applications to assess sources of nonlinearity in the dynamics of tested systems and/or in the measurement chain. Although coherence is not widely used in practice in conjunction with transmissibilities, the wavelet-based coherence can also be defined for wavelet-based transmissibility.

When measured functions are analyzed in the two-dimensional wavelet domain, the wavelet-based coherence can be defined as the ratio of the squared wavelet cross-power spectrum $G_{ij}$ and wavelet auto-power spectra $G_{ii}$ and $G_{jj}$ as [51]
(7)$$\gamma^{2}\left(a,b\right)=\frac{{\left|{\hat{G}}_{ij}\left(a,b\right)\right|}^{2}}{{\hat{G}}_{ii}\left(a,b\right){\hat{G}}_{jj}\left(a,b\right)}$$
where “^” indicates the averaging operator, when the subscript i indicates the response and the subscript j indicates the excitation, the wavelet-based coherence relates to the FRF. When these two subscripts indicate responses in different locations, the coherence given by Equation (7) relates to the transmissibility.

Coherence is a normalized measure and takes values between 0 and 1. In the modal analysis, the coherence function can indicate problems with estimated FRF. If the analyzed system is Linear Time-Invariant (LTI), and there are no problems with signal processing, coherence takes values close to one. Significant departures from a value of one indicate problems related to function estimation. Similarly, the transmissibility coherence checks the quality of the obtained transmissibility by checking the interrelation of the analyzed responses.

### 2.4. Numerical Implementation

The wavelet-based transmissibility defined by Equation (6) for two independent responses indexed i and j was implemented numerically for vibration analysis of time-variant systems. When such systems are analyzed it is impossible to average the time-frequency $T\left(a,b\right)$ estimate. This is due to the instantaneous nature of the analyzed system since its dynamics are not repeatable. Averaging is always performed to improve the signal-to-noise ratio. The work in [21,52,53] shows that various approaches—based on the crazy climbers algorithm, snake functions, and ridges/skeleton of the wavelet transform—can be used instead of averaging to smooth out the noise and enhance the results. However, this paper presents unenhanced results, assuming that the clarity and interpretation are not lost.

The wavelet-based coherence—given by Equation (7) and defined for two independent responses indexed i and j—was also implemented numerically. In contrast to the wavelet-based transmissibility, the wavelet-based coherence needs averaging to obtain an unbiased estimate. Without averaging the estimate of $\gamma^{2}\;\left(a,b\right)$ will be equal to one for the entire variable space. Another approach is used since averaging in the dual space domain is not possible—as explained in [20,21,51]—another approach is used. To average estimate of the wavelet-based coherence, the two-dimensional convolution functions are employed
(8)$${\hat{G}}_{ij}\left(a,b\right)=G_{ij}\left(a,b\right)*h\left(\tau_{1},\tau_{2}\right)=\int_{-\infty }^{+\infty }\int_{-\infty }^{+\infty }h\left(\tau_{1},\tau_{2}\right)G_{ij}\left(a-\tau_{1},b-\tau_{2}\right)d\tau_{1}d\tau_{2}$$
where $\tau_{1}$ and $\tau_{2}$ are the two-dimensional convolution time and scale operators, and $h\left(\tau_{1},\tau_{2}\right)$ is the two-dimensional convolution mask like a blurring box known from image processing. This convolution mask can be defined as [51]
(9)$$h=\frac{1}{n^{2}}\left[\begin{array}{ccc} 1 & \cdots & 1\\ \vdots & \ddots & \vdots \\ 1 & \cdots & 1 \end{array}\right]$$
where n is the length of each direction of the blurring box. Term $\frac{1}{n^{2}}$ is a normalization of amplitude concerning the size of the blurring box. The higher the n is, the bigger the blur radius is.

It is important to note again that when subscripts j and i indicate the excitation and the relevant response measurement, respectively, Equations (5) and (7) can be used to calculate the wavelet-based FRF and related wavelet wavelet-based FRF coherence, respectively. These two characteristics will also be used for comparative analysis. However, in contrast to FRFs, which are global characteristics, direct transmissibilities are local characteristics that depend on the location of measurement points.

## 3. Structural Damage Detection—Numerical Simulations

Numerical simulations are used to demonstrate how the wavelet-based transmissibility function can be used for damage detection. This section describes the simulated lumped-parameter time-variant system and presents damage detection results.

Firstly, the well-known concept of the so-called “frozen spectra”—i.e., classical frequency-domain characteristics calculated separately for all time values—was used to reveal the time-variant nature of the analyzed system. Then, the wavelet-based FRF and wavelet-based transmissibility are computed. The former is used for comparative analysis. The wavelet-based coherence is also used for both wavelet-based characteristics, i.e., the FRF and transmissibility.

### Lumped Parameter System

A simple 3-Degrees-of-Freedom (3-DOF) time-variant system—shown in Figure 1—was analyzed. The system consists of three masses connected by spring and viscous damping elements. The stiffness $k_{3}$ is time-variant and simulates the abrupt damage to the structure. Its value abruptly decreases from $k_{3}=960\;kN/m$ to $k_{3}=200\;kN/m$ at time t=5 s. All other physical parameters are constant. The values of these physical parameters are: $k_{1}=800\;kN/m$, $k_{2}=400\;kN/m$, $c_{1}=50\;N/\left(m/s\right)$, $c_{2}=50\;N/\left(m/s\right)$, $c_{3}=100\;N/\left(m/s\right)$, $m_{1}=7\;kg$, $m_{2}=7\;kg$, $m_{3}=10\;kg$. The mass $m_{2}$ was excited with white noise to simulate the real-life excitation. Responses were collected from the masses $m_{1}$ and $m_{3}$ as acceleration signals. The 3-DOF system was simulated in the Matlab/Simulink platform using the SimScape toolbox. The sampling frequency was equal to 1000 Hz in these simulations and the total time for the analyzed response was equal to 10 s.

Firstly, the analyzed 3-DOF system is simulated as a Linear Time-Invariant (LTI) to identify the reference values. In other words, two simulations with a constant value of $k_{3}$ are performed. The classical FRF and transmissibility with their corresponding coherence functions are calculated and are shown in Figure 2. For the FRF calculation, the response signal is taken from the mass $m_{3}$ and the excitation signal is taken from the mass $m_{2}$. For the transmissibility function calculation, two response signals are taken from masses $m_{3}$ and $m_{1}$.

For the 3-DOF system analyzed, the FRF functions in Figure 2a exhibit three resonances and two antiresonances. As expected, the resonances are shifted towards lower frequencies for the damaged structure. The amplitude of coherence functions in Figure 2b is reduced in FRF anti-resonances. The transmissibility in Figure 2c exhibits one major antiresonance that is shifted towards lower frequencies for the damaged system. Its coherence function (Figure 2d) shows similar behavior, i.e., the coherence function shows amplitude drops in frequencies related to antiresonances of the transmittance function. The relevant coherence’s amplitude reduction can be observed in Figure 2d for frequencies corresponding to this resonance. The frozen spectra results presented in Figure 2 reveal the dynamics of the undamaged and damaged simulated systems.

The simulated Linear Time-Variant (LTV) 3-DOF system was analyzed using the combined time-frequency domain. The wavelet-based FRF (input-output analysis) and transmissibility (output-only analysis) characteristics together with corresponding coherence functions, are shown in Figure 3. The results of all analyzed characteristics reveal the time-variant behavior of the system. The abrupt change of stiffness—leading to an increased value of natural frequencies—after 5 s can be observed in the FRF and the corresponding coherence function in Figure 3a,b, respectively. This abrupt change of stiffness—due to damage—can also be observed in the transmissibility and its coherence in Figure 3c,d, respectively. Two interesting observations can be made after this analysis. Firstly, the results in Figure 3 are blurred (or noisy) due to the lack of averaging, as explained in Section 2. Secondly, the wavelet-based transmissibility characteristics are less complicated (the coherence in particular) than the corresponding FRF characteristics that exhibit many resonances and antiresonances. Thus, the output-only analysis is easier to interpret and reveals the simulated abrupt damage more clearly.

## 4. Experimental Analysis—Measurements and Results

This section describes the experimental work undertaken to demonstrate damage detection based on output-only vibration measurements. Two damaged objects were investigated, i.e., a frame-like structural element and a two-storey (three floor) building model. Vibration responses were used to obtain the wavelet-based transmissibility and its coherence. The results were compared with the input-output vibration analysis based on time-frequency FRFs and relevant coherences.

### 4.1. Frame-like Structural Element

The first system investigated is a frame-like structural element shown in Figure 4. The 457 × 229 mm frame is welded from four 40 × 40 mm square steel bars. The SMC CD85N16-15-B pneumatic actuator is attached to both longer elements of the frame to provide initial tension force. This force is suddenly released during the experiment to simulate abrupt structural damage. When vibration tests are undertaken, the frame is excited using the electrodynamic The Modal Shop SmarkShaker K2004 shaker attached to the first of the longer elements. Band-limited white noise excitation signal is used as an input in these vibration tests to simulate real-life excitation. Vibration responses were gathered using the PCB 333B30 accelerometer attached to a second of the longer elements of the frame. The PCB 288D01 impedance head attached to end of the stinger and first of the longer elements was used to measure excitation force. The excitation and response data were acquired using the Polytec PSV-400 system. The input and output data were acquired using a 5120 Hz sampling rate. Recorder sequences were 10 s long and included 51,200 samples. Several measurements were taken to guarantee repeatability. However, data from only one representative measurement were used for damage detection due to the lack of averaging.

Altogether three different experimental tests were conducted. The structure was tested initially as a time-invariant system, i.e., the tension force applied by the actuator was constant, and input/output measurements—representing normal (undamaged) conditions—were taken. Then the structure was tested as a time-variant system, i.e., the applied tension force was released approximately 4 s after the test was initiated. Input and output data were taken for the entire period of the test. The third test was conducted for the frame that was not in tension (damaged condition), and the relevant input/output data were gathered. The analysis of time-invariant data and the wavelet-based input-output approach were used as a reference for the proposed wavelet-based analysis that utilized only output data.

The acquired data were analyzed using the same procedures described in Section 3. The classical vibration analysis—based on “frozen spectra”—was used initially to reveal the dynamics of the time-invariant undamaged and damaged system.

The classical FRF, transmissibility, and relevant coherence functions—corresponding to the first and third vibration test data—are given in Figure 5. The change of dynamics—due to damage—can be observed in the range 750 to 1250 Hz frequency band of the FRF in Figure 5a. The relevant coherence function—shown in Figure 5b—exhibits the reduction of amplitude for the points corresponding to resonances and antiresonances of the FRF. The only difference between the undamaged and damaged conditions can be seen at around 1050 Hz. The transmissibility and the relevant coherence—given in Figure 5c,d, respectively—reveal more changes for the damaged structures. The major differences in the analyzed characteristics can be observed in the range of 750 to 2500 Hz. Nevertheless, damage detection is difficult to detect when the classical input-output analysis is performed. The output-only analysis reveals some differences in the analyzed characteristics, but these differences are not significant to point out the tested damage. Environmental conditions could lead to similar changes in the analyzed characteristics, making the entire damage detection difficult, if not impossible.

The input/output data from the second test (time-variant structure) was then analyzed using the combined time-frequency analysis. The wavelet-based FRF (input-output analysis) and transmissibility (output-only analysis)—with their corresponding coherence functions—were calculated and are shown in Figure 6. This analysis is focused on the range 700 to 1700 Hz frequency band for the clarity of results. The results presented in Figure 6 show that both—i.e., input-output and output-only—characteristics indicate the change of the dynamics due to damage after 4 s. Even if the results are noisier and richer in terms of resonances and antiresonances, the output-only analysis, i.e., wavelet-based transmissibility and its coherence in Figure 6c,d, indicate this transition point in the dynamics as clearly, as the input-output analysis in Figure 6a,b. Damage detection this time is much easier than the analysis based on the classical results shown in Figure 5.

### 4.2. Three-Floor Building

The second experiment conducted in the presented damage detection analysis involves a simple model of a three-floor building, shown in Figure 7. The building model consisted of three plates simulating floors. These plates were connected by four continuous vertical rods. The top plate was additionally connected to the middle plate by a taut string (without any slack) to provide extra stiffness and stability to the structure. Similar to the sudden reduction of stiffness and/or loss of stability—for example, due to an earthquake—structural building damage was simulated by the sudden rupture of the taut string element.

The structure was excited when vibration tests were performed using the LDS PA1000L CE electrodynamic shaker attached to the base (i.e., the bottommost plate representing the bottom floor). Band-limited white noise excitation was used as an input in the experimental analysis to simulate the real-life excitation. The RDP DCT2000 transducer —attached to the bottommost plate—was used to gather the excitation. Vibration responses were measured using two Entran EGCS-A2-2 accelerometers attached to the middle and bottom floors. Although several tests were conducted to guarantee repeatability, representative data from only one test were used for damage detection. The input and output data were acquired using the NI SCXI-1321 with LabView 7.2 acquisition system. The sampling rate was equal to 200 Hz. Recorded signals were 60 s long, giving 12,000 data samples for each analyzed signal.

Three different experimental tests were conducted following the work reported in Section 4.1. Firstly, the structure was tested when the string between the middle and top floors was permanently fixed. That arrangement represented a time-invariant structure. Then the second test was conducted. The taut string was suddenly snapped 21 s after the test was initiated. The sudden change of stiffness represented the time-variant behavior. Finally, the structure with the snapped string was tested. That represented the time-invariant behavior, but in contrast to the first test, that behavior was analyzed for the structure with the permanently reduced stiffness.

Similar to the analysis shown in Section 4.1, the measurements taken for the two time-invariant tests (i.e., the first and the third test)—and the relevant results—were used as a reference for the time-variant analysis. In addition, the wavelet-based input-output analysis was performed for comparison with the proposed output-only analysis.

Once the excitation and response vibration data were gathered, the classical analysis—which utilized the time-invariant data from the first and third test—based on “frozen spectra” was performed. The classical FRF and transmissibility, together with the corresponding coherence functions, were calculated. The results—presented in Figure 8—show the dynamics of the structure with the initial and reduced stiffness.

As expected, structural damage can be detected by shifting the second resonance peak towards lower frequencies in Figure 8a. The antiresonance’s corresponding shift can also be observed in the transmissibility in Figure 8b. The relevant coherence functions in Figure 8c,d exhibit amplitude drops at frequencies corresponding to resonances and antiresonances. Although some shifts in these characteristics can be observed, the analysis is more difficult due to the rich nature of these resonances and antiresonances.

Finally, the analysis of the time-variant system—based on the second experimental vibration test—was performed. The wavelet-based FRF, transmissibility, and corresponding wavelet-based coherence were calculated. Figure 9 shows the results.

Structural damage can be detected very clearly by a shift of the natural frequency in the wavelet-based FRF in Figure 9a and a less significant shift of the antiresonance in the wavelet-based transmissibility in Figure 9c. It is important to note that the shift can be observed, and the instant of time at which these shifts happen can be identified to be 21 s. A similar observation can be made when the corresponding wavelet-based coherence is analyzed in Figure 9b,d. The wavelet-based transmissibility coherence is less complicated and thus easier for interpretation compared with the wavelet-based FRF coherence.

The two features observed in Figure 9 (i.e., the sudden frequency change (or shift) and its defined instantaneous location in the time domain) indicate that structural damage—not the environmental effect—is involved. This conclusion cannot be made reliably when the classical results are analyzed in Figure 8. Although both characteristics—i.e., wavelet-based FRF and wavelet-based transmissibility—indicate damage, the latter analysis requires output data only.

## 5. Conclusions

The wavelet-based transmissibility and its coherence were proposed to detect abrupt structural damage. Numerical simulations and experimental tests were used to demonstrate the method. The former utilized a simple 3-DOF structure exhibiting a sudden reduction of stiffness. The latter involved a simple frame element and a three-floor building model. The seeded damage in the experimental test was related to the sudden stiffness change. Damage detection results based on the output-only transmissibility were compared with the classical analysis based on “frozen spectra” and wavelet-based input-output FRF analysis.

The results presented in the paper lead to the following conclusions. The wavelet-based transmissibility can detect structural damage through a shift of the antiresonance. Similar behavior can be observed in the classical analysis based on FRF and transmissibility (frozen spectra) and in the wavelet-based FRF. However, only the combined time-frequency analysis based on wavelets (i.e., wavelet-based transmissibility and FRF) indicates the sudden change of the analyzed characteristics (i.e., resonances in FRF and antiresonances in transmissibility). The major advantage of the wavelet-based transmissibility is that—in contrast to the wavelet-based FRF—the method requires only output vibration data. This is particularly important when excitation measurements are not available. In addition to the wavelet-based FRF and transmissibility, the relevant coherences can also be used to indicate damage. The results show that the wavelet-based transmissibility coherence is less noisy (with fewer antiresonances) and easier to interpret than the wavelet-based FRF coherence.

In summary, the paper shows that wavelet-based transmissibility can detect abrupt changes in dynamics that can be associated with possible structural damage. However, the method is not superior to the wavelet-based FRF. However, its major advantage relates to the fact that only two output measurements (without excitation) are needed to detect damage, which is the real-life scenario, i.e., operational excitation forces are not measurable.

The presented combined time-frequency analysis’s major drawback is the lack of averaging, which leads to blurred time-frequency vibration characteristics. Any future work should address this important problem. More research is also needed with respect to the analysis of complex structures, various damage scenarios, and damage location problems.

## Figures and Tables

**Figure 1 materials-15-02722-f001:**
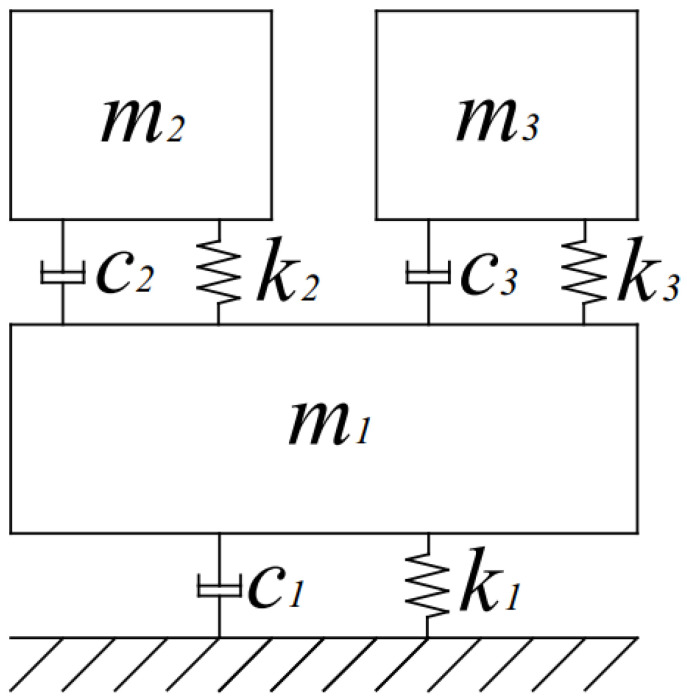
Schematic diagram of the simulated 3-DOF time-variant system.

**Figure 2 materials-15-02722-f002:**
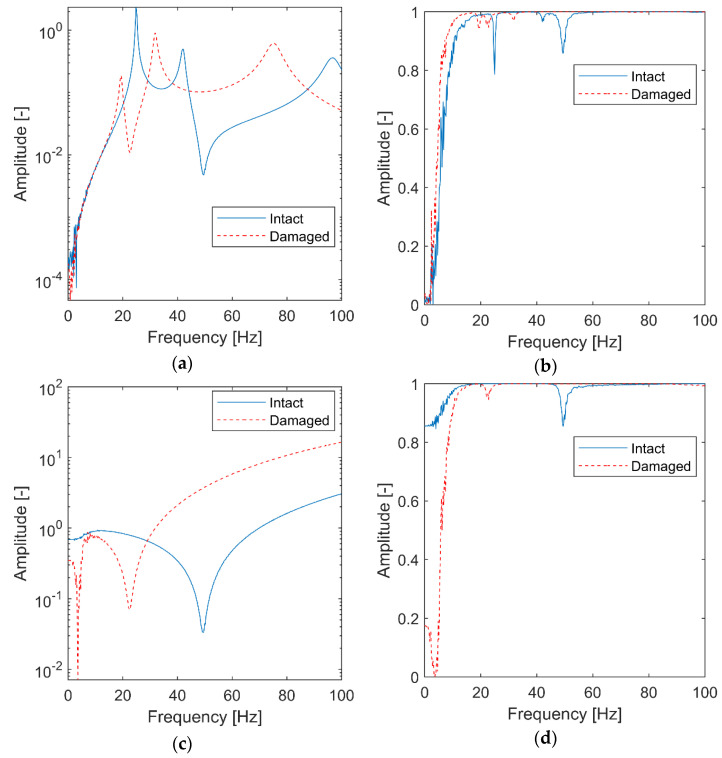
Classical analysis for the simulated LTI system: (**a**) FRF in a logarithmic scale; (**b**) FRF coherence; (**c**) transmissibility; (**d**) transmissibility coherence.

**Figure 3 materials-15-02722-f003:**
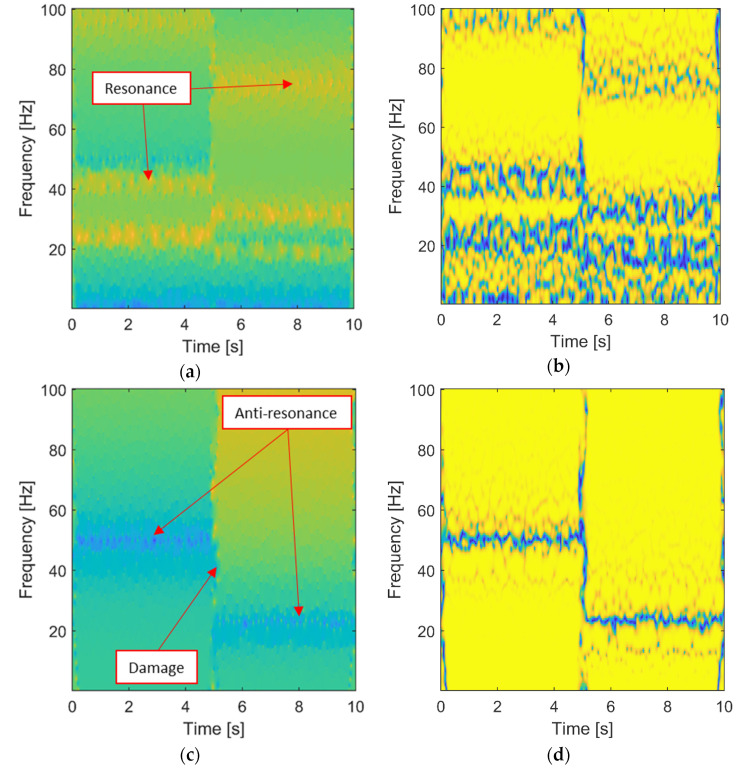
Wavelet-based analysis for the simulated LTV system: (**a**) wavelet-based FRF in logarithmic scale; (**b**) wavelet-based FRF coherence; (**c**) wavelet-based transmittance; (**d**) wavelet-based transmittance coherence.

**Figure 4 materials-15-02722-f004:**
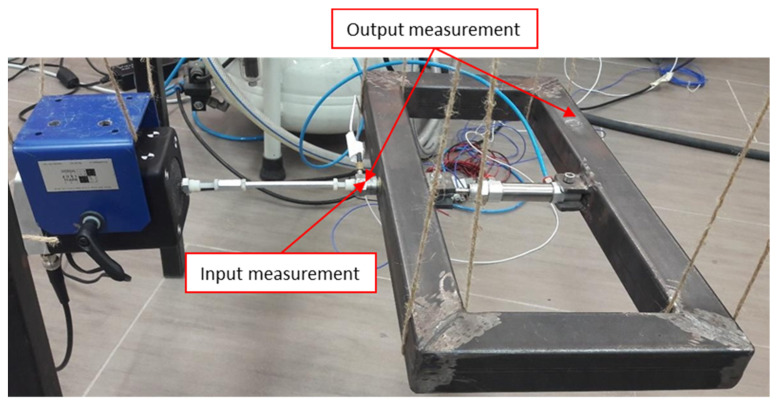
Frame structural element and experimental damage detection arrangements.

**Figure 5 materials-15-02722-f005:**
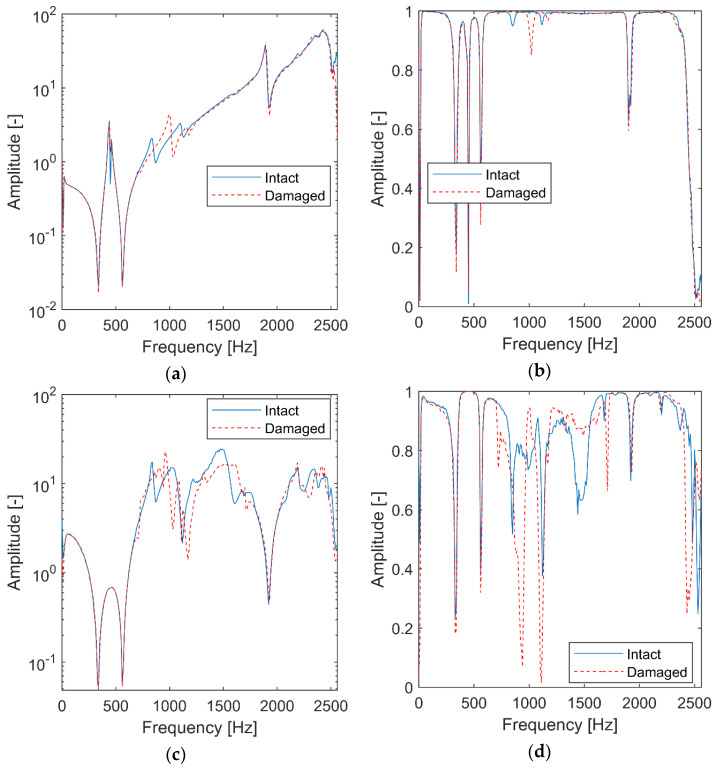
Classical analysis for the structural frame element: (**a**) FRF in a logarithmic scale; (**b**) FRF coherence; (**c**) transmissibility; (**d**) transmissibility coherence.

**Figure 6 materials-15-02722-f006:**
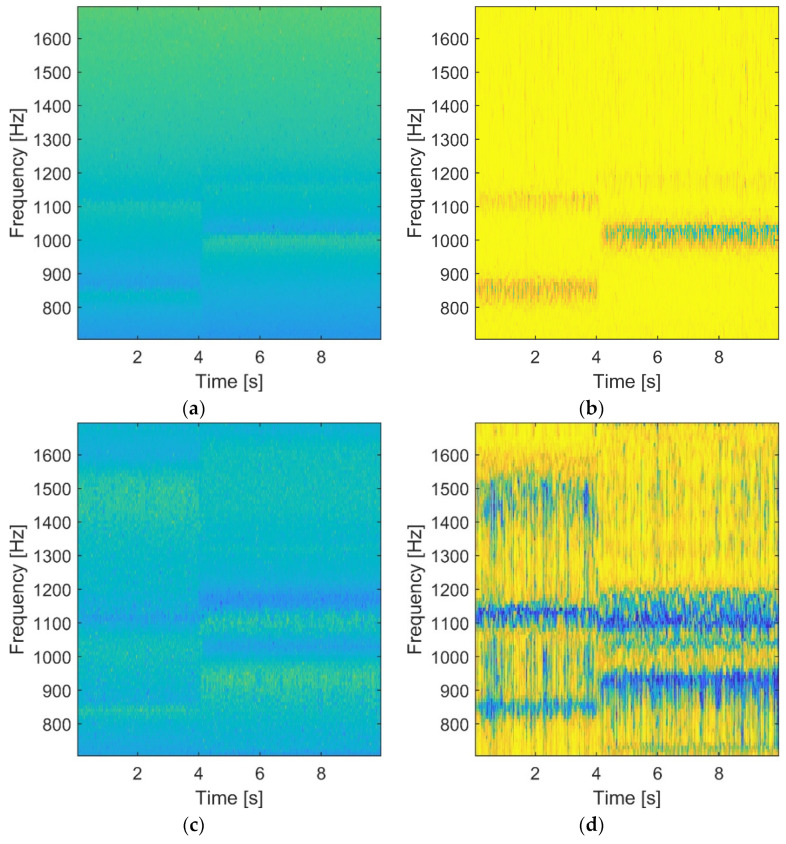
Wavelet-based analysis for the frame structural element: (**a**) wavelet-based FRF in a logarithmic scale; (**b**) wavelet-based FRF coherence; (**c**) wavelet-based transmissibility; (**d**) wavelet-based transmissibility coherence.

**Figure 7 materials-15-02722-f007:**
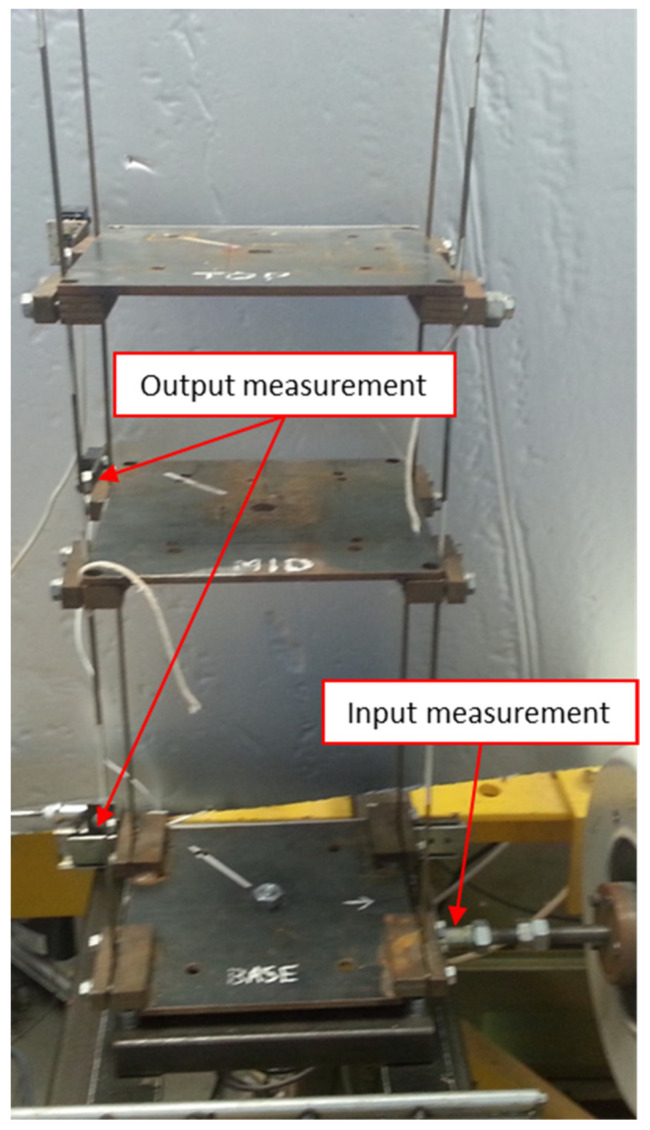
Three-floor building model and experimental set-up used for structural damage detection.

**Figure 8 materials-15-02722-f008:**
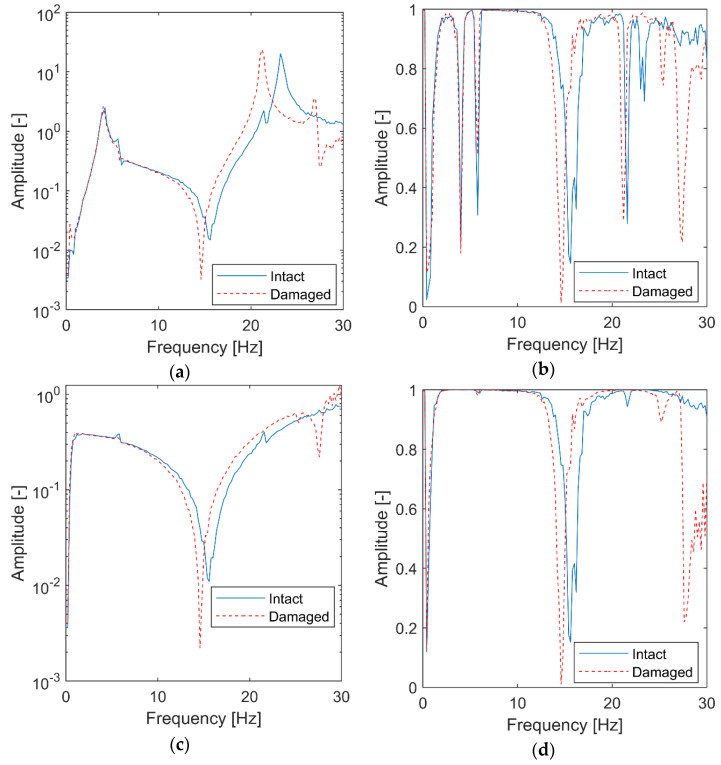
Classical analysis for the three-floor building: (**a**) FRF in a logarithmic scale; (**b**) FRF coherence; (**c**) transmissibility; (**d**) transmissibility coherence.

**Figure 9 materials-15-02722-f009:**
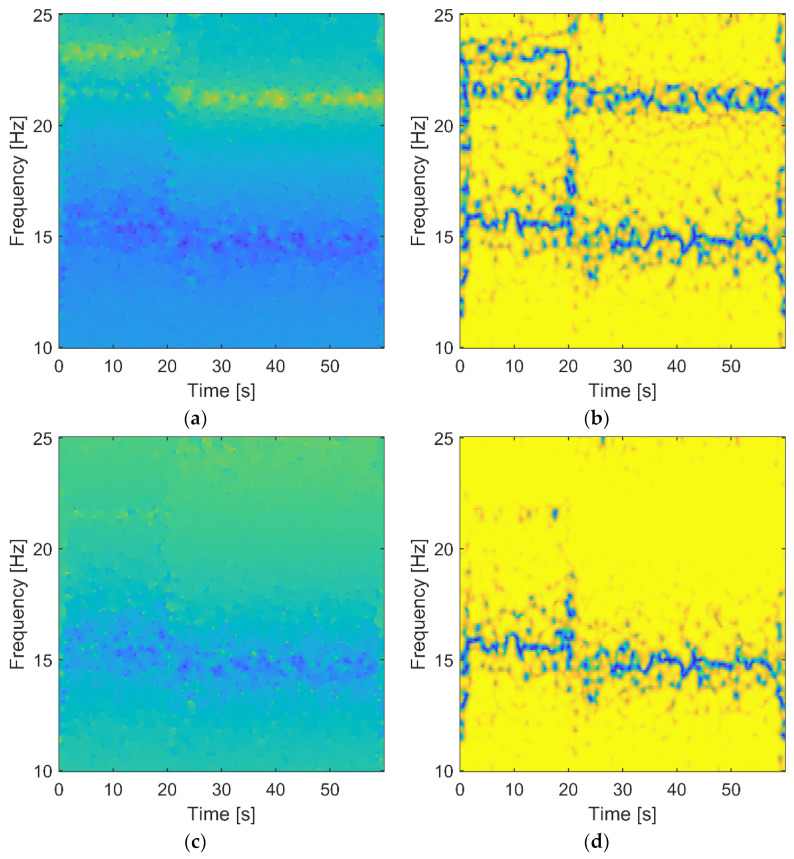
Wavelet-based analysis for the two-story building: (**a**) wavelet-based FRF in logarithmic scale; (**b**) wavelet-based FRF coherence; (**c**) wavelet-based transmittance; (**d**) wavelet-based transmittance coherence.

## Data Availability

Not applicable.

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
