# Peer review of "Wavelet-Based Transmissibility for Structural Damage Detection"

_materials, 2022, doi:10.3390/ma15082722_

Round 1

Reviewer 1 Report

The authors present a wavelet-based transmissibility study for structural damage detection. They first present characteristics of the observed wavelets and then perform a numerical analysis of the wavelets. The numerical results are compared with experimental measurements. The experiments are in good agreement with the predictions and structural damage can be detected clearly from the shift in frequency.

The paper is sound, interesting, and important for the detection of structural defects in buildings. It could be improved, in particular for clarity. For example, in eq. 1 x(t) is not defined, the word locality at the definition of a and b is unusual. Also, some symbols in equations are not written in italic. If the equations are described more clearly, the paper is suitable for publication.

Author Response

The authors present a wavelet-based transmissibility study for structural damage detection. They first present characteristics of the observed wavelets and then perform a numerical analysis of the wavelets. The numerical results are compared with experimental measurements. The experiments are in good agreement with the predictions and structural damage can be detected clearly from the shift in frequency.

The paper is sound, interesting, and important for the detection of structural defects in buildings. It could be improved, in particular for clarity. For example, in eq. 1 x(t) is not defined, the word locality at the definition of a and b is unusual. Also, some symbols in equations are not written in italic. If the equations are described more clearly, the paper is suitable for publication.

Description of x(t) is added in text, additionally locality is changed to location in the description.

Author Response

This paper reports impressive results on damage detection in mechanical structure using output-only wavelet-based transmissibility. The study background was extensively searched and suitably referred. I have several pointing outs and comments, which are as follows:

  1. The equation number and Figure number would not be matched in several points of paper main body, such as equation (6) and Figure 8.

Corrected things related to equation, but the parts related to Figure are ok, Figure 8 and 9 present same characteristics, but in time-invariant and time-variant scenarios and in text there are references of the two Figures to be compared.

  1. The frequency band would be difficult to understand in the first experiment. Used symbol would be mistaken.

This symbol is removed and descriptive text is added

  1. Please add explanation to equation (9) for helping reader’s understanding. In this equation, and  are not used in equation, and also “n” is not defined. Please explain the effect of this term, too.

Equation is updated and additional information concerning “n” is added in the text.

  1. The authors result in the conclusion as -based transmissibility coherence is less noisy and easier for The color map figures of the coherence in this paper would seem to be as this expression. But it is pointed out that there is an introduction of author decision in the color range of these time-variant figures. Please explain about the reason and the basis of the decision of color range in these coherence figures in three different experiments. Also add quantitative expression about the color range value which is introduced in each figure.

These are selected automatically by plotting program to cover entire range of amplitude. Authors do not think that they need to be explained, because this is standard procedure in data viewing, e.g., Figures related to classical 2D plots are also auto-ranged, Figures 2, 5 and 8, and this is straight forward how they were selected – in similar manner, i.e., min and max values were selected as ranges for color maps for Figures 3, 6 and 9.

  1. The main advantageous of the proposed method of this paper seems to be the expression “only two output measurements (without excitation)”, shown in Line 444 in 5. Conclusions. It should be modified in a suitable expression. Every experiment in this paper used an while noise excitation, and it would be indispensable, isn’t it? Also, please add comments about the while noise vibration source in the view point of actual use of this damage detection method.

This conclusion is extended in the paper, additionally comment on the white noise excitation has been added in text for all data sets.
